# A distinct p53 target gene set predicts for response to the selective p53–HDM2 inhibitor NVP-CGM097

Sébastien Jeay[1]*, Swann Gaulis[1], Stéphane Ferretti[1], Hans Bitter[2], Moriko Ito[1], Thérèse Valat[1], Masato Murakami[1], Stephan Ruetz[1], Daniel A Guthy[1], Caroline Rynn[3], Michael R Jensen[1], Marion Wiesmann[1], Joerg Kallen[4], Pascal Furet[5], François Gessier[5], Philipp Holzer[5], Keiichi Masuya[5†], Jens Würthner[6], Ensar Halilovic[7], Francesco Hofmann[1], William R Sellers[2], Diana Graus Porta[1]*

[1]Disease Area Oncology, Novartis Institutes for BioMedical Research, Basel, Switzerland; [2]Disease Area Oncology, Novartis Institutes for BioMedical Research, Cambridge, United States; [3]Metabolism and Pharmacokinetics, Novartis Institutes for BioMedical Research, Basel, Switzerland; [4]Center of Proteomic Chemistry, Novartis Institutes for BioMedical Research, Basel, Switzerland; [5]Global Discovery Chemistry, Novartis Institutes for BioMedical Research, Basel, Switzerland; [6]Translational Clinical Oncology, Novartis Institutes for BioMedical Research, Basel, Switzerland; [7]Translational Clinical Oncology, Novartis Institutes for BioMedical Research, Cambridge, United States

**Abstract** Biomarkers for patient selection are essential for the successful and rapid development of emerging targeted anti-cancer therapeutics. In this study, we report the discovery of a novel patient selection strategy for the p53–HDM2 inhibitor NVP-CGM097, currently under evaluation in clinical trials. By intersecting high-throughput cell line sensitivity data with genomic data, we have identified a gene expression signature consisting of 13 up-regulated genes that predicts for sensitivity to NVP-CGM097 in both cell lines and in patient-derived tumor xenograft models. Interestingly, these 13 genes are known p53 downstream target genes, suggesting that the identified gene signature reflects the presence of at least a partially activated p53 pathway in NVP-CGM097-sensitive tumors. Together, our findings provide evidence for the use of this newly identified predictive gene signature to refine the selection of patients with wild-type p53 tumors and increase the likelihood of response to treatment with p53–HDM2 inhibitors, such as NVP-CGM097.

**\*For correspondence:** sebastien. jeay@novartis.com (SJ); diana. graus_porta@novartis.com (DGP)

**Present address:** †Peptidream Inc, Tokyo, Japan

## Introduction

*TP53* is a tumor suppressor gene that functions to prevent cancer by allowing cells to recover from various stress insults such as DNA damage or by triggering their elimination when the extent of the damage is beyond repair. In its normal state, the p53 transcription factor acts in response to oncogenic or other stress signals to induce or repress a variety of target genes involved in cell cycle control, apoptosis, DNA repair, and cellular senescence (*Vogelstein et al., 2000*; *Harris and Levine, 2005*). In normal cells, the levels of p53 protein are tightly regulated by the E3 ubiquitin ligase HDM2 that targets p53 for ubiquitin-dependent proteasome degradation (*Haupt et al., 1997*; *Kubbutat et al., 1997*; *Marine and Lozano, 2010*). In addition, HDM2 binding to p53 blocks its

**eLife digest** Stress from daily activities and exposure to chemicals or UV radiation can all damage cells. Damaged cells may develop into cancerous tumors if unchecked. Normally, a protein called p53 helps to repair or eliminate damaged cells and prevent tumors from forming. The p53 protein does this by switching on or off genes that control DNA repair, cell division, and cell death. But half of all cancerous tumors have mutations that prevent p53 from doing its job.

Another protein called HDM2 keeps p53 in check by binding to p53 and preventing it from switching on and off genes after the stress passes. In cancers that have normal p53, sometimes HDM2 is overly active and prevents p53 from suppressing tumor formation and growth. Scientists are developing anticancer drugs that work by targeting HDM2; this frees p53 and allows it to wipe out cancerous cells. However, it is not always clear which patients with cancer are the most likely to benefit from anti-HDM2 therapy.

Jeay et al. screened hundreds of cancer cells to determine which ones are sensitive to HDM2-targeting drugs. As expected, the screen revealed that cancer cells that have mutations in the gene encoding p53 are insensitive to the anti-HDM2 drug because there is no working p53 to free up. But about 60% of the cancer cells that have normal p53 proteins also did not respond to the anti-HDM2 therapy. This finding indicates that the presence of normal p53 protein is necessary but not sufficient for tumor cells to respond to anti-HDM2 therapy.

Next, Jeay et al. compared the patterns of gene expression in the cancer cells that responded to an anti-HDM2 drug with those in cells that didn't respond. The analysis showed that a group of 13 genes are expressed more in the cells that responded to the drug. All 13 genes are unexpectedly direct targets of p53, suggesting that p53 remains active in these tumor cells, even if it is not working optimally. To verify these results, Jeay et al. grew human tumors in mice and found that tumors with high expression of the 13 genes are sensitive to the anti-HDM2 drug (called NVP-CGM097). The experiments strongly suggest that this 13-gene signature can be used to determine if a patient with cancer will respond to anti-HDM2 therapy. Following on from this work, researchers have already launched an early clinical trial with the anti-HDM2 drug and will test whether this gene signature is indeed useful in a real clinical setting.

transactivation domain preventing p53 transcriptional activation of its target genes (*Momand et al., 1992*). HDM2 is itself a p53 target gene and hence acts as part of a negative feedback loop which maintains low cellular concentrations of both partners under non-stressed conditions (*Picksley and Lane, 1993*; *Wu et al., 1993*; *Freedman et al., 1999*; *Michael and Oren, 2003*; *Bond et al., 2005*).

Approximately, 50% of all tumors display inactivating mutations in p53 (*Hainaut and Hollstein, 2000*) leading to its partial or complete loss of function (*Vogelstein et al., 2000*; *Levine and Oren, 2009*). In many cancers where *TP53* is not mutated, the function of the p53 pathway is often compromised through other mechanisms, including HDM2 gain of function by amplification and/or over-expression (*Bond et al., 2005*; *Vousden and Lane, 2007*; *Brown et al., 2009*; *Wade et al., 2010*). In these instances blocking the interaction between p53 and HDM2 is hypothesized to stabilize p53 leading to pathway activation and growth arrest and/or apoptosis in cancer. Based on this hypothesis and the structural elucidation of the p53–HDM2 interaction, several HDM2 small molecule inhibitors have been developed and are now in clinical trials. Indeed, prior work has shown that in human cancer cell lines or xenografts such inhibitors can elicit potent anti-tumor effects as a result of induction of cell cycle growth arrest and an apoptotic response (*Poyurovsky and Prives, 2006*; *Brown et al., 2009*; *Cheok et al., 2011*).

Here, we describe a novel and highly specific p53–HDM2 inhibitor, NVP-CGM097, currently in phase I clinical testing (NCT01760525) and a close analog NVP-CFC218 that are both based on an isoquinolinone scaffold. In order to identify patients most likely to respond to inhibitors of HDM2, we sought to develop patient selection biomarkers based on large-scale cancer cell line profiling. The analysis of sensitivity profiles across 356 cell lines led to the confirmation that p53 mutant cancer cells fail to respond to HDM2 inhibitors. However, among wild-type p53 cancer cells sensitivity was heterogeneous and not solely associated with HDM2 gene amplification. Using an unbiased

discovery approach, the expression of 13 genes (including HDM2) was found to have robust and superior predictive value for response compared to p53 wild-type status alone. This novel 13-gene signature was validated both in vitro and in vivo, and has the potential to improve the selection strategy of patients bearing p53 wild-type tumors who are most likely to respond to treatment with NVP-CGM097. Surprisingly, all 13 genes were p53 target genes suggesting that cells harboring at least partially activated p53 are those that are most susceptible to further p53 activation upon disruption of the p53–HDM2 interaction.

## Results

### Activity of NVP-CGM097 and NVP-CFC218 in biochemical and cellular assays

NVP-CGM097 and NVP-CFC218 are substituted 1,2-dihydroisoquinolinone derivatives that were designed to mimic three key hydrophobic interactions made by p53 residues Phe19, Trp23, and Leu26 in the HDM2 pocket (*Kussie et al., 1996*; *García-Echeverría et al., 2000*; *Furet et al., 2012*) (*Figure 1A*). The dihydroisoquinolinone core occupies the center of the cleft and allows for the positioning of appropriate substituents in this sub-pocket (manuscript in preparation).

The ability of NVP-CFC218 and NVP-CGM097 to disrupt the p53–HDM2 and p53–HDMX complexes was assessed in purified biochemical assays using time-resolved fluorescence resonance energy transfer (TR-FRET) to detect interactions between the N-terminal portion of HDM2 (amino acid 2 to 188) or HDMX (amino acid 2 to 185) and the human p53-derived peptide (amino acid 18 to 26). Both NVP-CFC218 and NVP-CGM097 displaced the p53 peptide from the surface of HDM2 with $IC_{50}$ values of 1.6 ± 0.2 nM and 1.7 ± 0.1 nM, respectively. In contrast, both compounds were substantially less active against HDMX with $IC_{50}$ values of 1300 ± 100 nM and 2000 ± 300 nM. These data show that NVP-CFC218 and NVP-CGM097 are specific inhibitors of the p53–HDM2 interaction (*Figure 1B*).

In measures of cellular viability, NVP-CFC218 and NVP-CGM097 elicited strong anti-proliferative effects in the HCT116 p53$^{WT}$ cell line and showed 34- and 35-fold selectivity, respectively, compared to an isogenic HCT116 cell line in which the *TP53* gene was deleted by homozygous recombination. Similarly, both compounds blocked proliferation of the osteosarcoma HDM2-amplified SJSA-1 cell line with 56- and 58-fold selectivity, respectively, compared to a control p53-null osteosarcoma SAOS-2 cell line (*Figure 1B*). These results indicate that NVP-CGM097 and NVP-CFC218 inhibit cell proliferation in a p53-dependent manner with a comparable potency and selectivity in vitro.

### Pharmacological activity of p53–HDM2 inhibitors in cancer cell lines

In order to investigate the activity of HDM2 inhibitors in preclinical models of cancer and to ultimately identify biomarkers predictive of response, we tested the anti-proliferative activity of NVP-CFC218 in a panel of 477 cell lines from the Cancer Cell Line Encyclopedia (CCLE) (*Barretina et al., 2012*). After quality control and manual curation of the dose response curves, a total of 356 cell lines met the cell viability quality criteria for NVP-CFC218 and were used for subsequent analyses. Cell lines were partitioned into sensitive and insensitive groups based on compound potency ($IC_{50}$). A rank-order plot of $IC_{50}$s for NVP-CFC218 showed a natural cut-off of 4 μM, which was used to categorize cell lines. Forty seven cell lines were categorized as sensitive and 309 cell lines as insensitive. For the sensitive cell lines, the maximal compound effect level (Amax) was ≤ −50% (*Figure 1C,D*, *Figure 1—source data 1*). As expected, based on the mechanism of action of NVP-CFC218, most of the sensitive cell lines harbored wild-type p53 (n = 43) and this association was statistically significant by Fisher's exact test (p-value = $5.1 \times 10^{-23}$) (*Figure 1E*). Moreover, repeat cell proliferation assays with the four sensitive cell lines bearing mutations in *TP53* showed them to be insensitive (P12-ICHI-KAWA, KBMC-2, and Hs 294T with $IC_{50}$ > 8 μM) or the reported mutation did not lead to complete p53 loss of function (NCI-H2122 carries both Q16L and C176F p53 mutations which are categorized as neutral and partially deleterious mutations by SIFT, respectively).

A parallel, unbiased orthogonal analysis of cell line sensitivities to over 2000 compounds, grouped by their mechanism of action (or main target), revealed that p53–HDM2 inhibitors including NVP-CFC218 were the most selectively enriched compound class in the p53 wild-type cell line set as

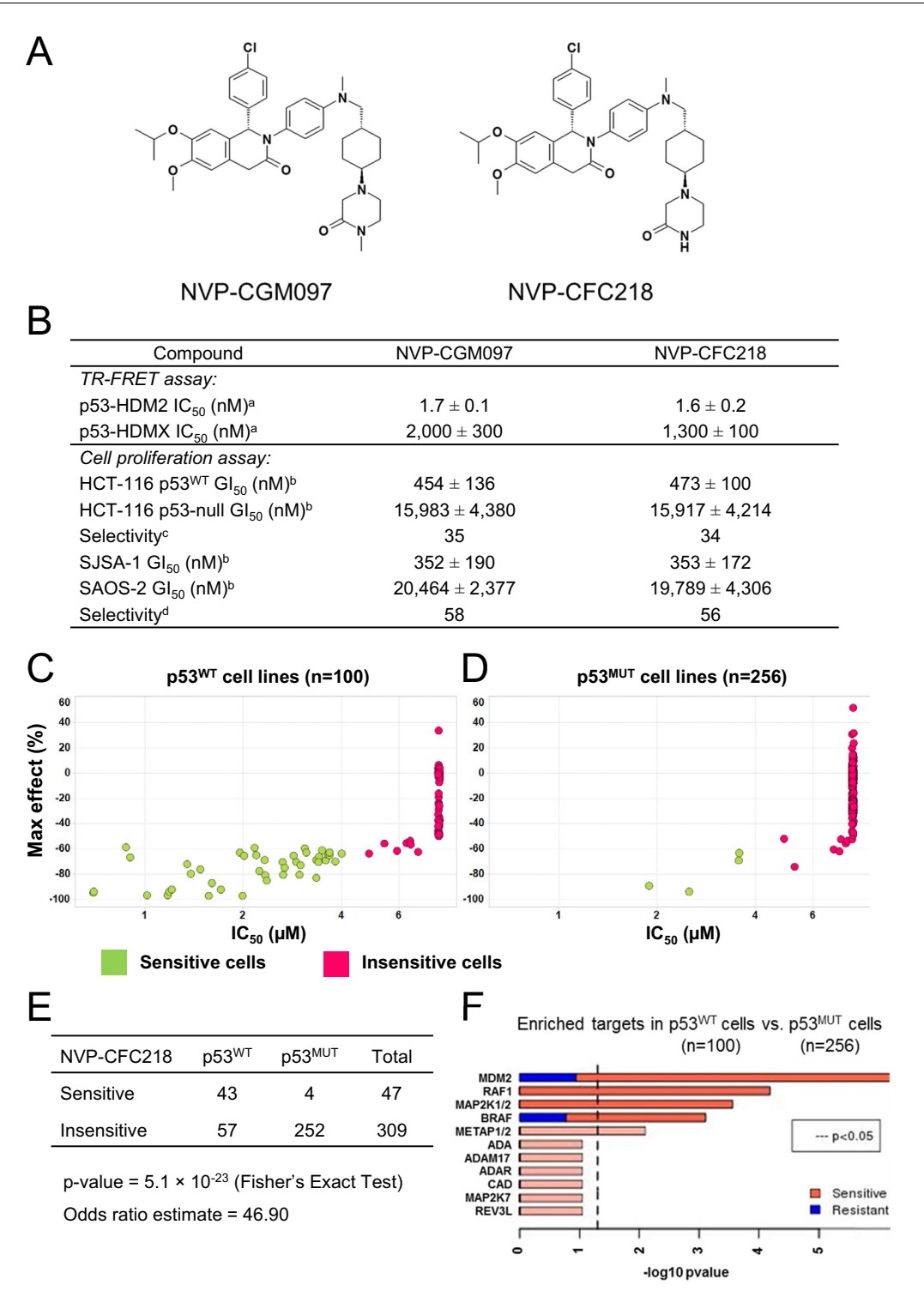

**Figure 1.** *TP53* wild-type status is necessary but not sufficient to predict sensitivity to NVP-CGM097 and NVP-CFC218. (**A**) Chemical structure of NVP-CGM097 and NVP-CFC218. (**B**) In vitro activity of NVP-CFC218 and NVP-CGM097 in TR-FRET binding assay ([a]) and cellular proliferation assay in human cancer cell lines ([b]). Data are expressed as concentration causing 50% inhibition and shown as mean ± SD from multiple (n ≥ 8) independent experiments. ([c]) Selectivity is determined by the ratio of $GI_{50}$ obtained using the HCT-116 p53-null and the HCT-116 p53[WT] isogenic pair of cell lines. ([d]) Selectivity is determined by the ratio of $GI_{50}$ obtained using SAOS-2 (p53-null) and SJSA-1 (p53[WT] and *HDM2*-amplified) osteosarcoma pair of cell lines. (**C and D**) Scatter plot showing $IC_{50}$ values expressed in μM of NVP-CFC218 in cell viability assays of p53 wild-type cell lines (**C**) and p53 mutated cell lines (**D**), colored by their response to NVP-CFC218. The data used to generate these plots, as well as cell line identity is available in *Figure 1—source*

*Figure 1 continued on next page*

*Figure 1 continued*

data 1. (E) Contingency table indicating the total number of sensitive and insensitive cell lines to NVP-CFC218. The p-value of $5.1 \times 10^{-23}$ shows a significant association between sensitivity to NVP-CFC218 and *TP53* wild-type status. (F) Main enriched compound target p-values from the Global Compound Selectivity Analysis. p-values are minus $\log_{10}$ transformed. The red color refers to compounds that are more selective in the wild-type p53 (p53$^{WT}$) strata than in the mutated p53 (p53$^{MUT}$) strata. The blue color indicates the reverse profile. Brighter colors indicate which target classes pass the 0.25 FDR cut-off. The length of each red/blue segment corresponds to the proportion of p53$^{WT}$ selective/p53$^{MUT}$ selective compounds in each target class.

The following source data is available for figure 1:

**Source data 1.** List of cell lines tested for their sensitivity to NVP-CFC218 (n = 356).

compared to the p53 mutant set, with both a false discovery rate (FDR) and a p-value below the cut-offs of 0.25 and 0.05, respectively (*Figure 1F*).

Interestingly, among all p53 wild-type cell lines for which compound data were available, 57% scored insensitive to NVP-CFC218 (*Figure 1C*). These results suggest that a patient selection strategy based only on the p53 status of the tumor will not optimally enrich for patients with a high likelihood of responding to this targeted therapy. Hence, there is a need for more sensitive and predictive biomarkers for p53–HDM2 inhibitors.

## Establishing a predictive model for p53–HDM2 inhibitor sensitivity in cell lines

We utilized a similar approach as described previously (*Barretina et al., 2012*) to identify molecular correlates of compound sensitivity to NVP-CFC218. We first applied a variance filter to remove half of the genes with the lowest variance, yielding 9053 genes. The group of sensitive cell lines (n = 47) was then compared to the group of most insensitive ones for which $IC_{50}$ and Amax of NVP-CFC218 was $\geq 8$ µM and $\leq -50\%$, respectively (n = 204). To build a predictive model, we used bootstrapping, whereby the cell lines were randomly split into training and testing sets. Within each bootstrapped data split where 2/3 of the data was used for training and 1/3 for testing, we used the Wilcoxon uni-variate test to select features that were significantly differentially expressed between sensitive and insensitive lines. Using the selected features, a naïve Bayes classifier was trained to predict sensitivity status. The number of features selected varied from 5 to 100 and for each, 20 bootstrapped data splits were conducted. We then assessed the classifier performance based on accuracy, sensitivity, and specificity of the predictions on the test data.

The classification accuracy, averaged over twenty bootstraps, was generally higher than 80% with a maximum of ~93% when 17 genes were selected (*Figure 2A*, red dot). The average classification accuracy with the fewest features within 1 SEM of the maximum was when 13 genes were selected. Class-level performance metrics, such as sensitivity and specificity, also showed that the 13-gene solution performed similarly to the optimum or optimally (*Figure 2—figure supplement 1*; *Figure 2—source data 1*). Sensitivity was the highest for the 13-gene solution, while the highest specificity was observed with 17 genes. However, the 13-gene solution was within 1 SEM of the best specificity. Thus, the 13-gene classifier was found to be the optimal solution, yielding the following performance statistics: 93% accuracy (±2.3%), 87% sensitivity (±10%), and 94% specificity (±2.8%).

After determining the number of genes to include in the model and assessing the prediction accuracy, we next wanted to select the 13 genes to include in our final model. To this end, we ran 100 bootstraps, selecting 13 genes based on the training set within each bootstrap, and tabulated the number of times each gene was selected. Fifty five different features were selected at least once and the top 13 features were selected more than 30 times. The 11 most frequent features were selected in more than 70 of the 100 bootstrapped selections, and the 9 most frequent ones appeared in more than 90 selections (*Figure 2B*).

The 13 top-ranking genes were strongly enriched in known p53 transcriptional target genes (*Table 1*). Specifically, 12 out of the 13 expression features were genes whose expression has been shown to be regulated by p53 (*Barak et al., 1993*; *el-Deiry et al., 1993*; *Miyashita et al., 1994*; *Okamoto and Beach, 1994*; *Varmeh-Ziaie et al., 1997*; *Tanaka et al., 2000*; *Adimoolam and Ford, 2002*; *Liu and Chen, 2002*; *Tan and Chu, 2002*; *Budanov et al., 2004*; *He and Sun, 2007*;

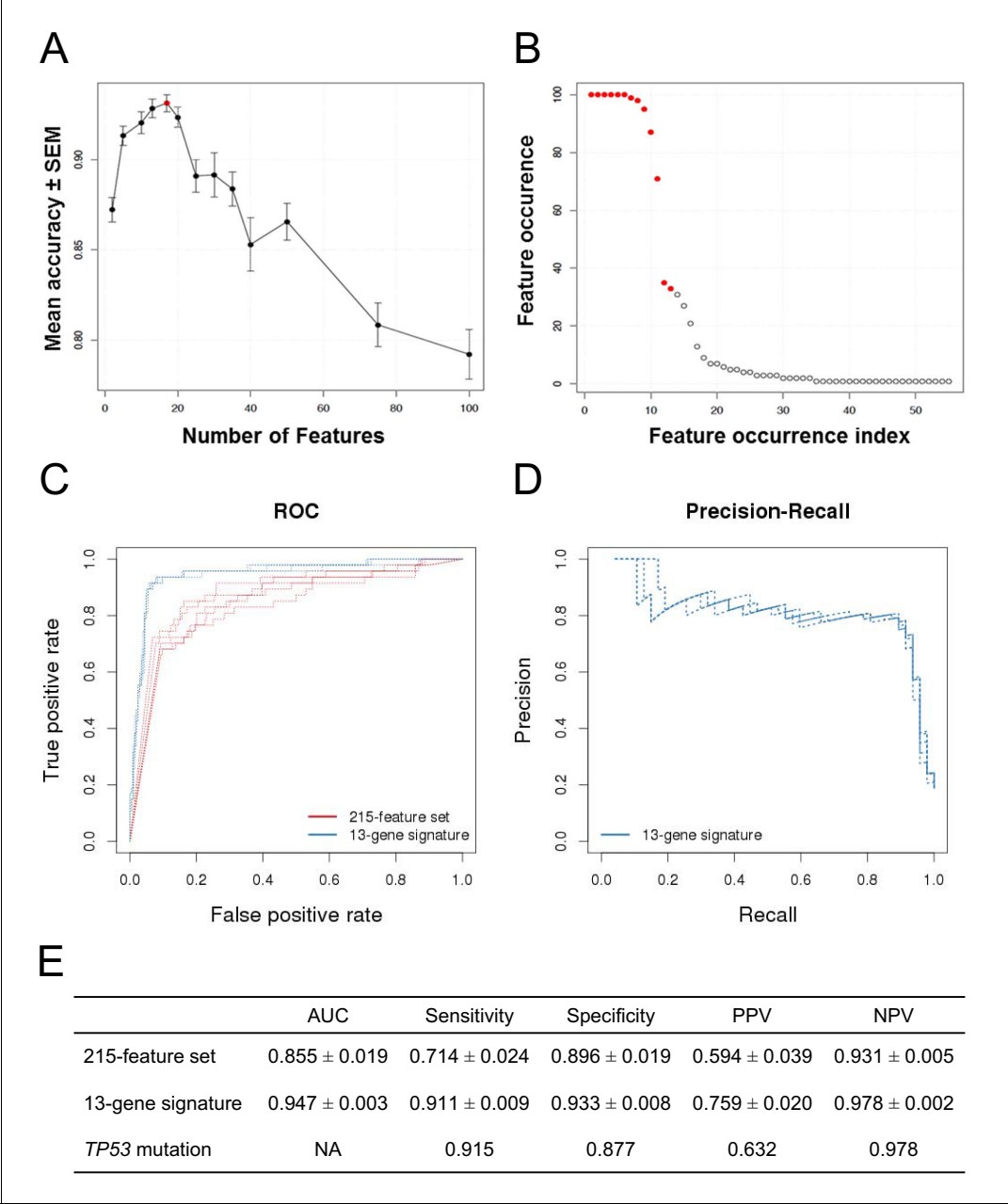

| | AUC | Sensitivity | Specificity | PPV | NPV |
|---|---|---|---|---|---|
| 215-feature set | 0.855 ± 0.019 | 0.714 ± 0.024 | 0.896 ± 0.019 | 0.594 ± 0.039 | 0.931 ± 0.005 |
| 13-gene signature | 0.947 ± 0.003 | 0.911 ± 0.009 | 0.933 ± 0.008 | 0.759 ± 0.020 | 0.978 ± 0.002 |
| *TP53* mutation | NA | 0.915 | 0.877 | 0.632 | 0.978 |

**Figure 2.** Predictive modeling of NVP-CFC218 chemical sensitivity from CCLE expression data. (**A**) Prediction accuracy estimation by bootstrapping analysis upon increasing feature set size. Estimates are averaged over 20 bootstrapping repeats. The maximum accuracy is observed for 17 features (red dot). Sensitivity and specificity are shown in *Figure 2—figure supplement 1*. The averaged data used to generate the accuracy, sensitivity and specificity plots is available in *Figure 2—source data 1*. (**B**) Feature occurrence frequencies of 13 feature selections in 100 bootstrapping repeats. Fifty five features were selected at least once. The 13 most frequently occurring features (red dots) were selected more than 30 times. (**C**) ROC curves for the three predictive models under comparison, i.e., the p53 mutation status, the 215-feature and the 13-gene signature models. (**D**) Precision-Recall plot for the 13-gene signature. Five curves are typically shown since cross-validation was repeated five times. (**E**) Performance estimates of the three compared predictive models: AUC (Area Under the Curve, from the ROC curve shown in **C**), Sensitivity (fraction of correctly predicted sensitive cell lines), Specificity (fraction of correctly predicted insensitive cell lines), PPV (positive predicted value, fraction of sensitive cell lines predicted as such), and NPV (negative predictive value, fraction of insensitive cell lines predicted as such). Measures are averaged over the 5 iterations of 5-fold cross-validations.

The following source data and figure supplements are available for figure 2:

**Source data 1.** Classifier performance with increasing feature set size.

**Figure supplement 1.** Prediction performance estimation by bootstrapping analysis upon increasing feature set size.

*Figure 2 continued on next page*

Figure 2 continued

**Figure supplement 2.** Prediction performance of NVP-CFC218 in p53[WT] CCLE cell lines.

*Kawase et al., 2008*; *Xiong et al., 2011*). Since no annotation for the 13[th] signature member was available at the time of the signature development, and to be consistent with the bootstrapping results, we replaced the 13[th] signature member by the 14[th] most frequently selected feature, TNFRSF10B, another p53 transcriptional target (*Wu et al., 1997*). Interestingly, all genes of the final gene signature were up-regulated in the sensitive cell lines (*Table 1*), implying that prior to compound treatment, a subset of p53 target genes is transcriptionally activated in the NVP-CFC218-sensitive cell lines.

## Comparing the 13-gene signature to p53 mutation at predicting p53–HDM2 inhibition sensitivity

After having initially restricted the analysis to gene expression, we evaluated all feature types, including copy number and mutation status in addition to gene expression. The same two-class comparison as before yielded a total of 215 significant features from multiple types. The *TP53* mutation status was the feature most associated with response to NVP-CFC218 (p-value = $2.46 \times 10^{-26}$), with an odds-ratio of 0.013.

To further evaluate the 13-gene signature model, we compared its prediction performance to that of both '*TP53* mutation status' and a naïve Bayes classifier trained on the '215-feature set'. We compared ROC (receiver operating characteristic) curves based on the predictions made on the test sets across 5 cross-validation runs for the models based on the 215-feature set, the 13-gene signature and the predictions given by *TP53* mutation status (i.e. the presence of a TP53 mutation predicts insensitivity, while a wild-type genotype predicts sensitivity). The AUC for the 13-gene signature was higher than for 215-feature set model: 0.947 vs 0.855 (*Figures 2C and E*). Furthermore, the precision–recall curves for the 13-gene signature showed that an 80% true sensitive prediction rate could be achieved while recalling more than 90% of the truly NVP-CFC218-sensitive cell lines (*Figures 2D and E*). The class-level performance measures showed that the 13-gene signature provided a substantial improvement in predicting response to NVP-CFC218 in particular when performance was evaluated by positive predictive value (PPV) (*Figure 2E*): 76% for the '13-gene

**Table 1.** 13-gene signature selected for the sensitivity prediction to p53–HDM2 inhibitors

| Features | Fold change | Wilcoxon p-value | Proposed function in p53 pathway |
|---|---|---|---|
| Expr. HDM2 | 2.18 | $3.09 \times 10^{-14}$ | Negative feedback loop |
| Expr. CDKN1A | 3.88 | $3.76 \times 10^{-11}$ | Cell cycle/senescence |
| Expr. ZMAT3 | 2.81 | $1.20 \times 10^{-10}$ | Positive feedback loop |
| Expr. DDB2 | 2.55 | $1.22 \times 10^{-10}$ | DNA repair |
| Expr. FDXR | 2.42 | $1.08 \times 10^{-09}$ | Apoptosis |
| Expr. RPS27L | 1.97 | $1.23 \times 10^{-09}$ | Positive feedback loop |
| Expr. BAX | 2.12 | $1.84 \times 10^{-09}$ | Apoptosis |
| Expr. RRM2B | 2.06 | $4.25 \times 10^{-09}$ | DNA repair |
| Expr. SESN1 | 2.27 | $1.04 \times 10^{-08}$ | Oxidative stress |
| Expr. CCNG1 | 1.69 | $4.81 \times 10^{-08}$ | Cell cycle |
| Expr. XPC | 1.62 | $1.56 \times 10^{-07}$ | DNA repair |
| Expr. TNFRSF10B | 1.91 | $3.30 \times 10^{-06}$ | Apoptosis |
| Expr. AEN | 1.46 | $3.30 \times 10^{-06}$ | Apoptosis |

Expr: gene-level expression values generated by the Affymetrix GeneChip technology with the HG-U133 plus 2 arrays, summarized according to the RMA normalization method.

signature' vs 59% for the '215-feature set' vs 63% for 'TP53 mutation status'. The enrichment of response when using the 13-gene signature and associated predictive model as a sample stratification strategy was also strongly significant when compared to the baseline response rate without prior stratification, estimated from the NVP-CFC218 sensitivity data as 19%.

Additionally in a p53 wild-type background (n=68), the 13-gene signature was compared to the p53 mutation model (*Figure 2—figure supplement 2*). Since in this comparison, the p53 model does not predict any insensitive cell lines, no AUC or NPV can be calculated. The table in *Figure 2— figure supplement 2C* shows a sensitivity of 94.9%, a specificity of 79.2%, a PPV of 88.7%, and a NPV of 90% for the 13-gene signature. In comparison, p53 mutation model has a sensitivity of 100%, a specificity of 0%, and a PPV of 63.2%. As before, the 13-gene model outperforms the p53 mutation model in terms of PPV (*Figure 2—figure supplement 2C*).

Thus, these results show that the 13-gene signature provides an improvement in response prediction to drug treatment over both the *TP53* mutation status and the larger signature consisting of 215 significant features. Moreover, these results suggest that the 13-gene signature has utility in TP53 wild-type genetic backgrounds.

## Predicting HDM2 inhibitor sensitivity in independent cell lines and primary tumors

The predictive value of the 13-gene signature was subsequently tested in a second, independent data set of cell lines representative of multiple cancer types representative of the lineages included in the first proliferation screen. Cells were assayed for their sensitivity to both p53–HDM2 inhibitors NVP-CGM097 (n = 52) and NVP-CFC218 (n = 38) in in vitro proliferation assays. Because of the manual assay format differing slightly from the high-throughput assay, the cut-off for sensitivity was fixed at $IC_{50} \leq 3$ µM for both p53–HDM2 inhibitors and predictions of sensitivity for every cell line were derived using the 13-gene signature. As expected, cells that were indeed sensitive or insensitive to NVP-CGM097 were similarly sensitive or insensitive to NVP-CFC218, except for one cell line (COLO-829) (*Figure 3—source data 1*).

Overall, 36/52 and 26/38 cell lines were predicted to be sensitive to NVP-CGM097 and NVP-CFC218, respectively, while 27/52 and 23/38 were truly sensitive (*Figure 3A,B*). While we observed a slight decrease in specificity relative to the bootstrap estimate, sensitivity measures were 89% and 91% (*Figure 3C*), respectively, which is comparable to the sensitivity estimate obtained by bootstrapping the predictive model training data (87.3% ± 10.4%, see *Figure 2—source data 1*). Noticeably, PPV values for both drugs were higher than basal response rate (*Figure 3C*). Taken together these data validate the 13-gene signature as a robust predictor for sensitivity to both NVP-CGM097 and NVP-CFC218 p53–HDM2 inhibitors in an external sample set. We also determined the performance of the 13-gene signature in pre-selected cells based upon the presence of wild-type p53. As shown in *Figure 3—figure supplement 1*, 35/42 and 25/30 wild-type p53 cell lines were predicted to be sensitive to NVP-CGM097 and NVP-CFC218, respectively, while 25/35 and 21/25 were truly sensitive. PPV values were moderately higher than the basal response rate by 5 to 7%, and the relatively small number of insensitive p53^WT cell lines in this external set greatly impaired the specificity and NPV values (*Figure 3—figure supplement 1C*). Taken together, these results indicate the 13-gene signature to be a better predictive feature than *TP53* mutation status in unselected cell lines. In addition, our results in both the discovery (*Figure 2*) and the validation sets (*Figure 3*) suggest the 13-gene signature may improve the response rate in wild-type p53 pre-selected cell lines .

Given the role of p53 in responding to cellular stress, an obvious concern would be that this signature might simply be reflective of cells undergoing cellular stress during in vitro culture. In order to exclude this possibility, we assessed the presence of the signature in primary human tumor samples. For this purpose, we used the 13-gene signature, associated with the naïve Bayes predictive model trained on cell line data, to interrogate a collection of ~21,300 human tumor samples for which the whole genome expression profiles are available. Here, we reasoned that the proportion of samples of any given lineage that scored positively for the signature would be a measure of the concordance between cell line expression data and primary human tumor data. The specific proportion of any given lineage among primary human tumor samples scoring positive for the presence of the 13-gene signature (*Figure 4A*) was found to be very similar to the lineage proportions of signature positive, sensitive cell lines in the CCLE (*Figure 4B*), as probed by the NVP-CFC218 chemical sensitivity experiment described above. These data suggest that the 13-gene signature is not likely an

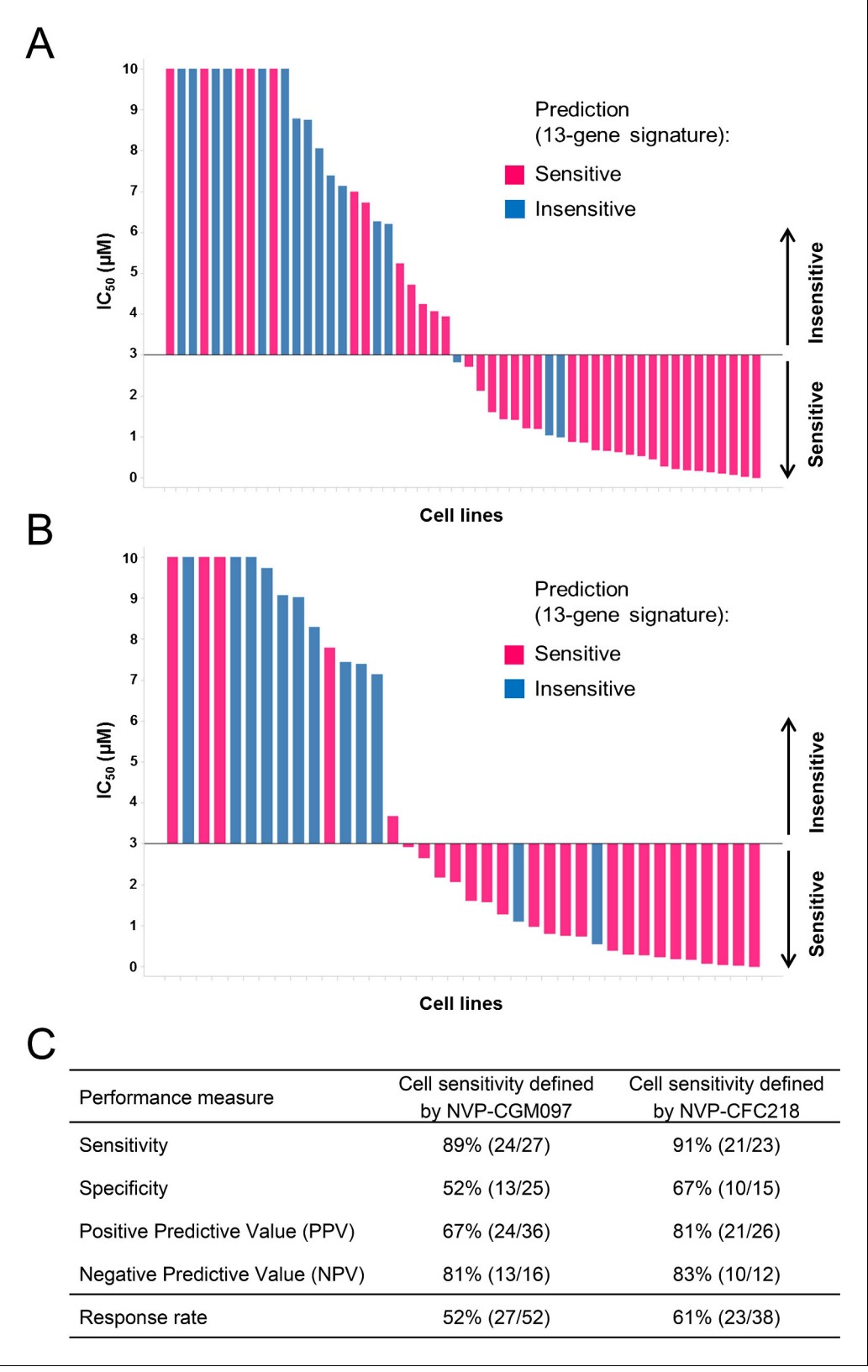

**Figure 3.** Validation of the predictive 13-gene signature in an external set of cell lines. A total of 52 or 38 cell lines were tested against NVP-CGM097 (**A**) or NVP-CFC218 (**B**), respectively, in a 3-day proliferation assay. Sensitivity call for each compound was applied according to a cut-off of 3 μM. Cells that were predicted as sensitive are

*Figure 3 continued on next page*

*Figure 3 continued*
shown in pink, while the ones predicted as insensitive are shown in blue. The data used to generate the waterfall plots and cell line details are available in *Figure 3—source data 1*. Performance values of the 13-gene signature were derived from this experiment, and are shown in (C).
The following source data and figure supplement are available for figure 3:

**Source data 1.** Sensitivity prediction and sensitivity to NVP-CFC218 and NVP-CGM097 of an external set of cell lines (n = 52).
**Figure supplement 1.** Validation of the predictive 13-gene signature in p53^WT cell lines.

artifact of the in vitro culture. In keeping with this observation, the 13-gene signature identified a fraction of predicted sensitive human primary tumor samples within our patient-derived tumor xenograft (PDX) collection (n = 503) (*Figure 4C*), thus allowing the selection of models for further in vivo validation of its predictive power. In summary, these analyses provide an estimation of the prevalence of 13-gene signature positivity across various cancer indications and underscore additional cancer types with high signature positivity, such as hepatocellular carcinoma and renal cell carcinoma that may have been underrepresented in the in vitro cell line profiling approach.

## Predicting sensitivity to NVP-CGM097 in patient-derived tumor xenograft models

To further assess the predictive performance of the 13-gene signature, tumor response in vivo was tested across a set of PDX models from indications identified from the predictions described above (*Figure 4C*).

These in vivo experiments were performed with NVP-CGM097, owing to its superior pharmacokinetic properties and oral bioavailability as compared to NVP-CFC218 (*Table 2*). First, the pharmacodynamic effects and optimal daily dose of NVP-CGM097 were determined in the cell line-derived SJSA-1 osteosarcoma xenograft model that harbors an amplification of the *HDM2* gene. A single oral dose of NVP-CGM097, 100 mg/kg, led to stabilization of p53 and elevation of p53 target genes such as *CDKN1A* (p21) at the mRNA level (*Figure 5A*, left) and at the protein level (*Figure 5A*, right). Treatment of SJSA-1 xenografted tumors with NVP-CGM097 led to dose-dependent tumor growth inhibition with 65% tumor regression at 100 mg/kg daily (*Figure 5B*, left) and was well tolerated as measured by body weight (*Figure 5B*, right). The anti-tumor activity of NVP-CGM097 correlated well with a dose-dependent induction in tumors of p21 and HDM2 at the mRNA level and/or protein levels (*Figure 5C,D*). Thus, based on these results, the dose and schedule chosen for the follow up in vivo validation of the 13-gene signature was established at 100 mg/kg NVP-CGM097 given orally, once daily.

A total of 55 PDX models were used for these studies. NVP-CGM097 sensitivity was predicted for this sample set using the 13-gene signature (see 'Materials and Methods' section for the specificities of these predictions and *Figure 6—source data 2*). As shown in *Figure 6A* and *Figure 6—source data 1*, 27/55 human primary xenograft tumor models were predicted to be sensitive to NVP-CGM097, and of these 27 models, 19 were truly sensitive, resulting in a PPV of 70%, improving the basal response rate of 49% (*Figure 6C*). Moreover, 28/55 in vivo models were predicted to be insensitive, and 20/28 were found to indeed be truly insensitive, leading to a significant NPV of 71% (*Figure 6C*). Also, Sensitivity and Specificity features were comparable to the predictive model performance estimated on cell lines by the bootstrapping protocol (*Figure 6C*; *Figure 2—source data 1*). We also determined the performance of the 13-gene signature in tumors pre-selected based upon the presence of wild-type p53. As shown in *Figure 6B*, 21/33 wild-type p53 human PDX models were predicted to be sensitive to NVP-CGM097 and of these, 18 were truly sensitive, resulting in a PPV of 86%, again significantly improving the basal response rate of 70% (*Figure 6C*). In addition, 12/33 models were predicted to be insensitive to the drug, and 7/12 were found to be truly insensitive, leading to a significant NPV of 58% (*Figure 6C*).

In conclusion, the in vitro predictive model performance shown in *Figure 2* and *Figure 2—source data 1* was confirmed in vivo using PDX models. The results further validate the 13-gene signature

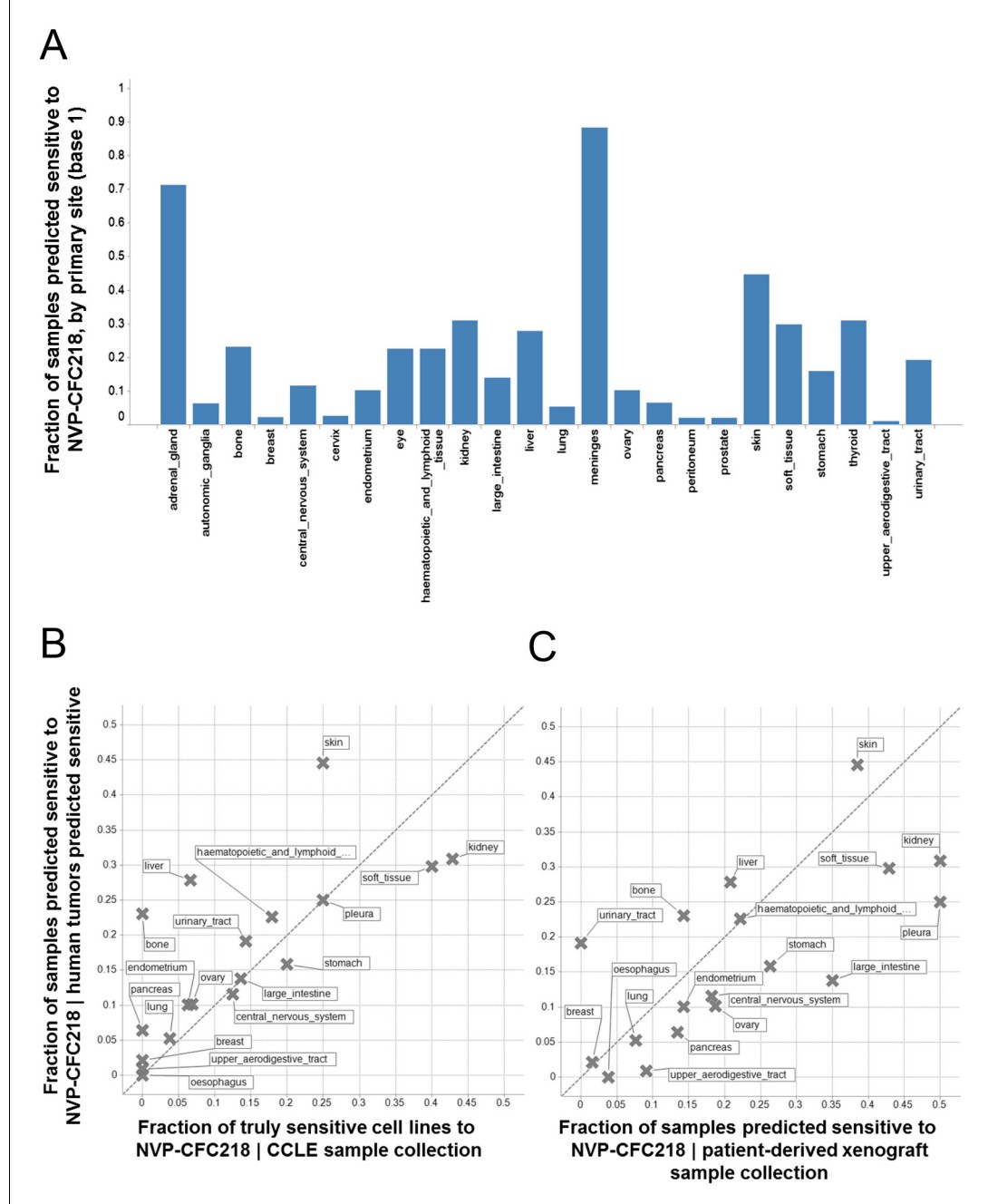

**Figure 4.** Prevalence of the 13-gene signature in human primary tumors and patient-derived tumor xenograft models. (**A**) Fraction of human primary tumors predicted sensitive by the 13-gene signature. Samples are grouped by primary site. Expression data from 21,300 human primary tumors were compiled and only primary sites consisting of more than 50 samples were included in the analysis. The primary site nomenclature is based on the COSMIC Classification System. (**B**) Correlation between the fraction of human primary tumor samples predicted sensitive and the observed sensitivity ratio in the CCLE cell line data. (**C**) Correlation between the sensitivity predictions from the human primary tumor sample collection and the patient-derived tumor xenograft model collection. Human primary tumor samples, patient-derived tumor xenografts, and CCLE cell lines are organized by lineage. The dashed line corresponds to the identity line.

as a better predictor for response to p53–HDM2 inhibitors as compared to a selection strategy solely based on p53 mutation status. Notably, the findings described here support the use of p53–HDM2 inhibitor sensitivity predictive model in either non-selected tumors or wild-type p53 pre-selected tumors.

## Discussion

In this study, we have identified a novel 13-gene signature that robustly predicts sensitivity to p53–HDM2 inhibitors, such as NVP-CGM097. This novel 13-gene signature has a superior predictive value as compared to p53 wild-type status alone and importantly, when applied to p53 wild-type preselected tumors, it provides a further enrichment of the drug response rate.

The newly discovered close analogs, NVP-CGM097 and NVP-CFC218 (*Figure 1A*), are both members of a novel chemical class (dihydroisoquinolinones) which differs from Nutlin analogs (e.g., Nutlin 3a and RO5045337/RG7112) or other p53–HDM2 inhibitors currently being tested in the clinic (e.g., RO5503781/RG7388, SAR405838/MI-773, AMG 232, DS3032b) (*Masuya et al., 2014*, manuscript in preparation). As to compare with Nutlin 3a, NVP-CGM097 and NVP-CFC218 show differences in binding mode within the p53 binding site of HDM2 (*Jeay et al., 2014*; *Valat et al., 2014*, manuscript in preparation), which allows better in vitro and in vivo on-target potency, more favorable drug-like properties, and improved in vivo behavior of this compound family (*Ferretti et al., 2014*; *Jeay et al., 2014*). NVP-CGM097 and NVP-CFC218 were first tested in cell-free assays, where both were shown to selectively inhibit p53–HDM2 binding. In cells, consistent with their mechanism of action (*Cheok et al., 2011*) and in line with their highly selective nature, both compounds showed comparable anti-proliferation effects in HCT-116 and SJSA-1 wild-type p53-expressing cell lines. The isogenic HCT-116 p53-null cells and the osteosarcoma SAOS-2 p53 null cells remained insensitive to NVP-CGM097 and NVP-CFC218 with $IC_{50}$s $\geq$ 10 µM. Furthermore, in cell viability assays, using 356 cancer cell lines from the CCLE representative of various tumor types (*Barretina et al., 2012*), NVP-CFC218 was shown to inhibit proliferation of about 13.2% of the cancer cell lines tested at relevant concentrations (*Figure 1B*). As expected, 91.5% (43/47) of NVP-CFC218-sensitive cells expressed wild-type p53, consistent with its mechanism of action (*Figure 1C,E*). Importantly, the sensitivity of 3/4 of these cells bearing mutations in *TP53* couldn't be reproduced (P12-ICHIKAWA, KBMC-2 and Hs 294T with $IC_{50}$ > 8 µM), while the sensitivity of the NCI-H2122 *TP53* mutant cell line was

**Table 2.** PK parameters of NVP-CFC218 and NVP-CGM097 in preclinical species

| PK parameters | Mouse | Rat | Dog | Monkey |
|---|---|---|---|---|
| NVP-CFC218 | | | | |
| CL (mL/min.kg) | 9 | 12 ± 1 | 3 ± 1 | 11 ± 2 |
| Vss (L/kg) | 3.1 | 8.0 ± 1.8 | 3.6 ± 0.5 | 2.1 ± 0.3 |
| $t_{1/2app.}$ (h) | 3.6 | 9.3 ± 1.9 | 14.4 ± 0.7 | 2.6 ± 0.3 |
| $AUC_{inf}$ (µM.h) i.v.* | 2.80 | 2.21 ± 0.21 | 8.08 ± 1.68 | 2.37 ± 0.31 |
| $AUC_{inf}$ (µM.h) p.o.* | 1.20 | 1.07 ± 0.34 | 5.74 ± 0.68 | 0.34 ± 0.05 |
| Cmax (µM) p.o.* | 0.094 | 0.075 ± 0.023 | 0.240 ± 0.043 | 0.077 ± 0.014 |
| Tmax p.o. (h) | 2.0 | 4.0 ± 1.4 | 3.3 ± 1.2 | 1.0 ± 0.1 |
| Oral BA (%F) | 43 | 49 ± 15 | 71 ± 8 | 14 ± 2 |
| NVP-CGM097 | | | | |
| CL (mL/min.kg) | 5 | 7 ± 1 | 3 ± 1 | 4 ± 1 |
| Vss (L/kg) | 2.6 | 6.4 ± 0.4 | 3.8 ± 0.4 | 2.0 ± 0.4 |
| $t_{1/2app.}$ (h) | 6.4 | 12.1 ± 1.1 | 14.2 ± 1.8 | 8.3 ± 0.9 |
| $AUC_{inf}$ (µM.h) i.v.* | 4.70 | 3.66 ± 0.27 | 9.44 ± 1.43 | 6.59 ± 0.74 |
| $AUC_{inf}$ (µM.h) p.o.* | 3.34 | 2.97 ± 0.40 | 7.08 ± 1.05 | 3.73 ± 0.78 |
| Cmax (µM) p.o.* | 0.207 | 0.134 ± 0.016 | 0.250 ± 0.030 | 0.363 ± 0.020 |
| Tmax p.o. (h) | 4.0 | 4.5 ± 1.9 | 2.7 ± 1.2 | 1.3 ± 0.6 |
| Oral BA (%F) | 71 | 81 ± 11 | 75 ± 11 | 57 ± 12 |

Mouse (n = 1 per time point) and rat (n = 3 per time point) were treated either with a single dose of the indicated compound at 1 mg/kg i.v. or 3 mg/kg p.o. Dog and monkey (n = 3 per time-point) were treated either with a single dose of the indicated compound at 0.1 mg/kg i.v. or 0.3 mg/kg p.o.
*AUC and Cmax values are shown dose-normalized to 1 mg/kg.

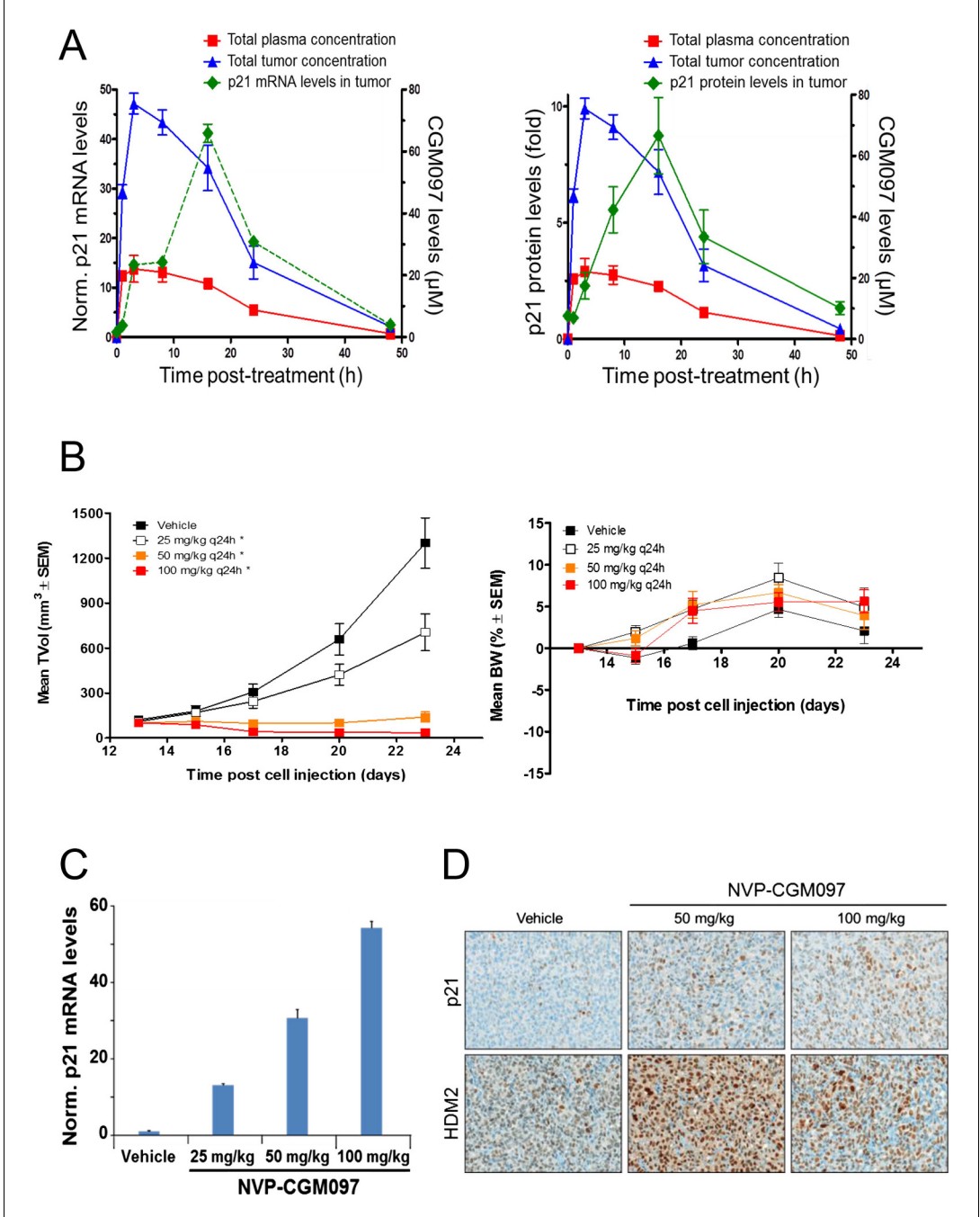

**Figure 5.** Activity of NVP-CGM097 in SJSA-1 tumor-bearing mice. (**A**) PK/PD relationship. Mice (n = 3/time-point) bearing established subcutaneous SJSA-1 xenografts received vehicle or a single oral dose of 100 mg/kg NVP-CGM097. Levels of NVP-CGM097 were measured in plasma and in tumor at 7 different time-points within 48 hr following the dose. As a pharmacodynamic readout, the p53 target gene p21 mRNA (left panel) and protein (right panel) levels were assessed in the tumor samples by qRT-PCR and reverse phase protein array at 7 time-points within 48 hr following NVP-CGM097 single dose. Data are plotted as mean ± SEM. (**B**) In vivo anti-tumor activity of NVP-CGM097. Mice (n = 6/dosing group) bearing established subcutaneous SJSA-1 xenografts received vehicle or 25, 50, or 100 mg/kg of an oral suspension of NVP-CGM097 daily. Tumor volumes were calipered throughout the study, and data are plotted as mean ± SEM (left panel). The body-weight (BW) is expressed as the percentage change in BW relative to day 0 (initiation of treatment). Data are plotted as mean ± SEM (right panel). *, p < 0.05 vs vehicle control. (**C** and **D**) Tumor pharmacodynamics effects of NVP-CGM097. Tumor samples were retrieved 3 hr post-last dose at the end of the efficacy study described in (**B**). Tumor p21 mRNA levels were assessed by qRT-PCR and data are plotted as mean ± SEM (**C**). Tumor samples were retrieved and paraffin-embedded 3 hr post-last dose at the end of the efficacy experiment described in (**B**). p21 and HDM2 protein levels were assessed by immunohistochemistry (**D**).

confirmed in repeat cell proliferation assays. Interestingly, the reported p53 mutations in NCI-H2122 cells (i.e., Q16L and C176F p53) are categorized as being only partially deleterious, suggesting the p53 pathway remains partially functional in these cells. In line with the strong association of p53 mutation with lack of sensitivity to NVP-CFC218, sensitivity response comparison to over 2000 compounds, grouped by their mechanism of action, showed that wild-type p53 cell lines in the CCLE were most sensitive to p53–HDM2 inhibitors, including NVP-CFC218 (*Figure 1F*).

Interestingly, 57% (57/100) of the p53 wild-type cell lines for which compound sensitivity data were available, scored insensitive to NVP-CFC218 (*Figure 1C and E*). These results indicate that the selection of tumors based only on their p53 mutation status may not be optimal in order to tailor patient selection towards maximal response rate. With the aim to discover more sensitive biomarkers for patient selection, an in vitro predictive model was developed based on the integrative analysis of the genomic features of this panel of 356 cancer cell lines and their association with sensitivity to NVP-CFC218 (*Barretina et al., 2012*). A 13-gene signature was found to be the optimal solution at predicting p53–HDM2 inhibition from gene expression (*Figure 2*). A further outcome of these studies identified p53 mutation status as being a strong predictor for insensitivity to NVP-CFC218 among about 40,000 input features containing genomic, lineage, and gene expression features, confirming in an unbiased manner the underlying hypothesis for this specific mechanism of action. In addition, a broader set of 215 newly discovered significant features were identified as being correlated to NVP-CFC218 sensitivity. Interestingly, HDM2 expression was found as the second ranked significant feature (after p53 mutation) confirming its correlation with sensitivity to p53–HDM2 inhibitors in an unbiased manner. However, HDM2 amplification was not revealed by the modeling, most likely because of the low representation of HDM2-amplified cell lines in CCLE. Surprisingly, 13/14 of the top most significant expression features corresponded to well-known p53 target genes and their up-regulated expression was found associated with increased sensitivity to NVP-CFC218 (*Table 1*). The 14th feature, unannotated at the time of this work, was found to be RPL22L1, previously described as having a central role in αβ T lymphocytes development, together with its highly homologous paralog RPL22, by up-regulating p53 in a tissue-specific manner (*Anderson et al., 2007*; *Zhang et al., 2013*). The positive predictive value of this 13-gene signature was found to outperform both the p53 mutation feature alone and the 215 significant feature-set (76% vs 63% and 59%, respectively) (*Figure 2E*). Furthermore, comparable performances were found in an additional set of independent cell lines tested for their sensitivity to NVP-CGM097 and NVP-CFC218, validating the predictive power of the 13-gene signature in vitro (*Figure 3*).

This cell line-derived 13-gene signature was demonstrated to be suitable for the identification of human primary tumors predicted to be sensitive to a p53–HDM2 inhibitor across a set of about 21,300 human tumor samples and across a collection of 503 patient-derived tumor xenograft models, with available genome-wide expression data. Interestingly, the prevalence of the 13-gene signature was found to be quite variable within the multiple lineages tested (*Figure 4A*). Importantly, however, the lineage-specific proportion of samples scoring positive for the 13-gene signature was strikingly comparable to the proportion of sensitive cell lines to NVP-CFC218, in a given lineage. Furthermore, the application of the signature across a large set of human tumor samples revealed new indications of potential interest for the development of p53–HDM2 inhibitors, such as renal cell carcinoma and hepatocellular carcinoma (*Figure 4C*), that had been previously missed due to their under-representation in the set of CCLE cell lines tested. The ability of the 13-gene signature to differentiate signature-positive from signature-negative patient-derived tumor xenografts, allowed testing its in vivo predictive value across a set of these models. In line with the in vitro findings, the 13-gene signature was found to greatly outperform the random response rate to NVP-CGM097 in vivo, independently upon whether the xenograft models were pre-selected or not for their p53 mutation status (*Figure 6*; *Figure 6—source data 1*).

Many research efforts have been devoted to the characterization of molecular mechanisms conferring gene-specific regulation within the p53 network (*Vousden and Prives, 2009*). It has been observed that in proliferating cells, minimal activity of p53 can lead to basal expression of some of its target genes, such as p21 (*Tang et al., 1998*), through mechanisms involving basal promoter-specific recruitment of transcription initiation components (*Espinosa and Emerson, 2001*; *Espinosa et al., 2003*). In a more recent study from *Allen et al. (2014)*, using Global Run-On sequencing, gene-specific regulatory mechanisms affecting a number of key survival and apoptotic genes within the p53 network were uncovered. The authors imply that some p53 target genes can

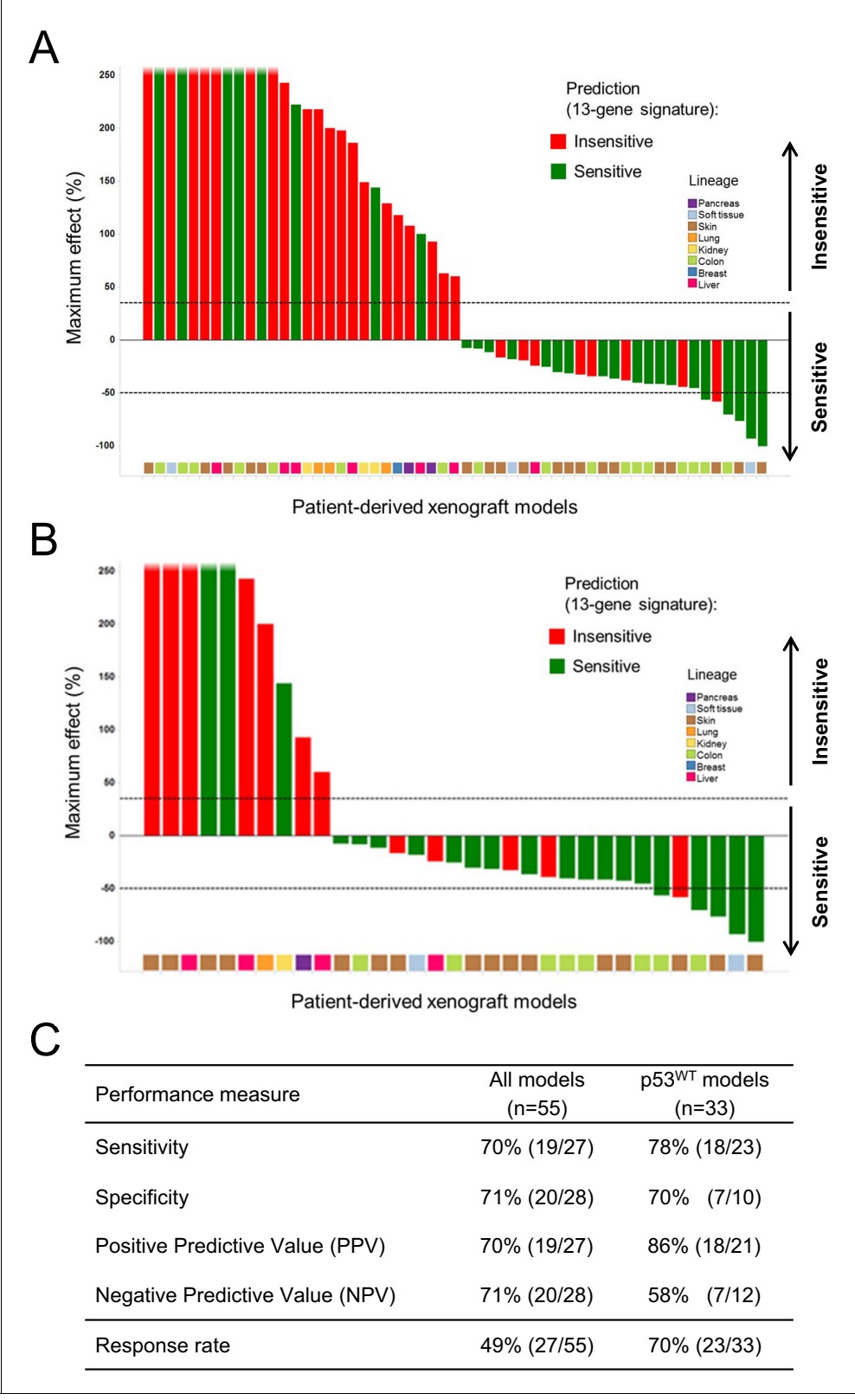

**Figure 6.** Validation of the predictive 13-gene signature in patient-derived tumor xenografts (PDX). PDX models were established by subcutaneous implantation in nude mice of surgical tumor tissues from treatment-naive cancer patients. 55 PDX models of multiple cancer types were used for the study: melanoma (n = 19), colorectal cancer (n = 17), liposarcoma (n = 3), renal cell carcinoma (n = 3), hepatocellular carcinoma (n = 7), breast cancer (n = 1), pancreatic cancer (n = 2), and lung cancer (n = 3). Tumor-bearing mice (n = 4/dosing group/model) received vehicle or 100 mg/kg of NVP-
*Figure 6 continued on next page*

*Figure 6 continued*

CGM097 daily for 4 weeks. The response is reported as percentage change in tumor volume at a given day of treatment relative to day 0 (start of treatment). The cut-off used for sensitivity to NVP-CGM097 was based on RECIST adapted for full tumor volume measurement. The sensitivity call was made at the maximum effect time-point during the treatment period. Sensitivity predictions for each PDX model were generated using the 13-gene signature (*Figure 6—source data 2*). (A) Waterfall plot showing the tumor response to NVP-CGM097 for all the PDX models in the study (n = 55), color-coded by the prediction output. (B) The same visualization as in (A) is shown restricted to the wild-type p53 PDX models (n = 33). The data used to generate the waterfall plots and PDX model details are available in *Figure 6—source data 1*. Performance values of the 13-gene signature were derived from this experiment and are shown in (C).

The following source data is available for figure 6:

**Source data 1.** Sensitivity prediction and sensitivity to NVP-CGM097 of a set of in vivo PDX models (n = 55).

**Source data 2.** Expression data of each of the 13 genes included in the gene signature in the set of in vivo PDX models (n = 55).

be 'primed' to be switched on even before the p53 protein is activated, leading to a rapid transcription of these genes following p53 activation (*Allen et al., 2014*). Some of these p53 target genes, such as *CDKN1A*, *HDM2*, *CCNG1*, *DDB2*, *FDXR*, *BAX*, *TNFRSF10B*, and *AEN*, were found to be within the top most transcribed genes following Nutlin-3 treatment (*Allen et al., 2014*) and coincidently are all included in the 13-gene signature described in *Table 1*. Taking together, these observations strongly suggest that the p53 target gene set contained within the predictive 13-gene signature is representative of an underlying activated p53 pathway that renders cancer cell lines and patient-derived tumor xenografts sensitive to p53–HDM2 inhibitors. Thus, it is reasonable to think that the 13-gene signature discovered here is best at predicting sensitivity to p53–HDM2 inhibitors, since it not only discriminates mutant p53 from wild-type p53 tumors, but also tumors that have an intact p53 gene but a silent p53 pathway from those that are p53 wild-type and bear an activated pathway. We postulate that the latter may be more prone to quickly induce a robust p53-dependent transcriptional response upon exposure to p53–HDM2 inhibitors, thus being more sensitive to this specific anti-cancer therapy.

Altogether, this work highlights the power of a biomarker discovery approach based on large-scale data generation and unbiased data analysis, followed by robust validation using various independent samples sets. Furthermore, the finding that the 13-gene signature corresponds to a set of p53 target genes and that its positivity reflects activity of the p53 pathway, demonstrates the intimate relationship between the biomarker set and the mechanism of action of the targeted therapy, hence its potentially broad applicability to any selective p53–HDM2 inhibitor for optimal patient selection. On these bases, we are currently evaluating the 13-gene signature in a Phase I clinical trial (NCT01760525), in cancer patients bearing p53 wild-type tumors treated with NVP-CGM097.

## Materials and methods

### Chemical entities

NVP-CFC218 and NVP-CGM097 were synthesized by Global Discovery Chemistry (NIBR, Novartis, Basel, Switzerland). For in vitro studies, 10 mM stock solutions were prepared in 100% dimethyl sulfoxide (DMSO). For in vivo experiments, NVP-CGM097 was formulated immediately before each oral administration in a suspension of 0.5% hydroxy-propyl-methyl-cellulose (HPMC).

### Biochemical assay

NVP-CFC218 and NVP-CGM097 biochemical activity against the p53–HDM2 and p53–HDMX interactions was assessed in a TR-FRET assay. Standard assay conditions consisted of 60 μl total in PBS buffer containing 125 mM NaCl, 0.001% Novexin, 0.01% Gelatin, 0.2% Pluronic F-127, 1 mM DTT, and 1.7% final DMSO. Both NVP-CFC218 and NVP-CGM097 were added at different concentrations to 0.1 nM biotinylated HDM2 (amino acids 2-188) or 0.1 nM HDMX (amino acids 2-185), 0.1 nM Europium-labeled streptavidin and 10 nM Cy5-p53 peptide (Cy5-p53 amino acids 18–26). After incubation at room temperature for 30 min, samples were measured on a GeniosPro reader (Tecan). FRET assay readout was calculated from the raw data of the two distinct fluorescence signals

measured in time resolved mode (fluorescence 665 nm/fluorescence 620 nm × 1000). IC$_{50}$ values were calculated by curve fitting using XLfit (Fit Model #205).

## Cell lines and in vitro pharmacologic cell line profiling

The isogenic HCT-116 p53-null cell line was obtained from Horizon (Cambridge, UK). All other cell lines were obtained from ATCC (American Type Culture Collection), DSMZ (Deutsche Sammlung von Mikroorganismen und Zellkulturen), and HSRRB (Health Science Research Resources Bank) and cultured in RPMI or Dulbecco's modified Eagle's medium plus 10% FBS (Invitrogen, Carlsbad, CA) at 37°C, 5% CO$_2$. Cell line identities were confirmed using a 48-variant SNP panel. Effects of NVP-CFC218 and NVP-CGM097 on cellular growth and loss of viability shown in *Figure 1B* were measured using the YOPRO assay (Invitrogen) and IC$_{50}$ values were calculated by curve fitting using XLfit (Fit Model #201). A detailed description of the high-throughput cell viability assays can be found in the report of *Barretina et al. (2012)*.

## Global compound selectivity analysis

To identify compounds to which wild-type p53 cell lines were selectively responsive as compared to non-wild-type p53 cell lines across the CCLE, we used the high-throughput cell line profiling described above and reported in *Barretina et al. (2012)*. First, a selected compound sensitivity metric was log2 transformed and a selectivity score was computed by multiplying the metric Z-score by the absolute value of the metric. We then performed a Wilcoxon test opposing the two groups of cell lines and corrected for multiple testing using Benjamini and Hochberg FDR, followed by an enrichment analysis by the compound's main target. To this end, compounds were grouped into target classes, and Fisher's exact tests were performed with compounds considered as significantly differentially responsive when the FDR was below 0.25. The cut-off for the FDR is lenient as this is used for enrichment purposes. Finally, we corrected the p-values obtained from enrichment analysis for multiple testing.

## Genomic and genetic characterization of CCLE cell lines

The genomic and genetic characterization of a panel of cancer-relevant cell lines was undertaken mostly as described (*Barretina et al., 2012*). Briefly, gene-level expression values were generated with the HG-U133 plus 2 array (Affymetrix GeneChip technology) and summarized according to the RMA normalization method; gene-level chromosome copy number values were obtained with the Affymetrix SNP6.0 technology and processed using the Affymetrix apt software and expressed as log2 transformed ratios to a collection of HapMap reference normal samples; gene-level genetic alterations (point-mutations, insertions, deletions, and complex alterations) were compiled from the Sanger center COSMIC data and internal sources including Exome Capture Sequencing of 1600 cancer-related genes; pathway-level expression values were generated as referenced in the GeneGo Metacore knowledge base; cell line lineage (cell line tissue of origin); gene-level 'tumor suppressor status' was generated by integrating the genetic alteration, copy number, and expression information. Such genomic and genetic data were assembled in a matrix that covers a total of about 40,000 genomic features and were used to test their association with cell line chemical sensitivity to NVP-CFC218.

## Affymetrix GeneChip RMA normalization

All Affymetrix GeneChip Human Genome U133 Plus 2.0 arrays were pre-processed with a custom chip definition file (Entrez Gene-centric custom CDF version 16 from the University of Michigan) using the RMA (robust multi-array average) summarization/normalization algorithm (*Irizarry et al., 2003*; *Dai et al., 2005*). A set of pre-computed reference quantiles and probe effects was input to the algorithm for normalization of Affymetrix arrays in an approach known as the Extrapolation Strategy or refRMA (*Goldstein, 2006*; *Katz et al., 2007*). The reference quantiles and probe effects were defined from a compendium of publically available arrays representing a broad diversity of oncology indications (through the expO data set but not exclusively) and human body normal tissues (GEO series accessions GSE5764, GSE6338, GSE2109, GSE28504, GSE6764, GSE7307, GSE32317, GSE7753, GSE8507, GSE7904, GSE8671, GSE8762, GSE4107, GSE4183, GSE8977,

GSE20238, GSE20596, GSE13911, GSE10282, GSE9829, GSE9843, GSE7553, GSE9891, GSE9899, GSE6004 and GSE4237). The RefPlus R package was used to that aim (*Harbron et al., 2007*).

## Predictive modeling of p53–HDM2 inhibition chemical sensitivity in vitro

To define the most accurate and parsimonious predictive model from CCLE expression data, a bootstrapping protocol was used first which is described in the 'Results' section. All calculations were done with R-3.0.2 using principally e1071 and caret R libraries.

The broader, 215-feature set, that discriminates NVP-CFC218 sensitive from insensitive cell lines, was obtained as follows: Wilcoxon signed-rank tests or Fisher's exact tests were used to compare the genomic features of the sensitive to insensitive groups. Features having continuous values (gene expression and copy number, pathway expression features) were subjected to Wilcoxon sum rank test, while those having discrete values (genetic alteration, tumor suppressor status, and lineage features) to Fisher's exact test. Significant features passed a local (by feature type) FDR of 0.25. Irrespective of the FDR, a minimum or maximum number of features per feature type were also required. To minimize the impact of the high degree of correlation among the features on the feature selection step, the feature data were clustered before statistical testing by Affinity Propagation method (*Frey and Dueck, 2007*) to only keep features representing the most variability.

To compare sensitivity prediction accuracies of the 13-gene signature to both, the 215-feature set and *TP53* mutation, naïve Bayes probabilistic models were then built from the respective feature sets. The performances of classifications were evaluated with 5 repeats of 5-fold cross-validations of the train data. To transform the naive Bayes probabilities into a sensitive or insensitive class-level prediction, a threshold was defined as the probability maximizing the sensitivity and specificity. The entire and nearly identical procedure is described in more details in *Barretina et al. (2012)*.

## Predictive modeling of p53–HDM2 inhibition chemical sensitivity of human primary tumors and patient-derived tumor xenograft models

The human primary tumor sample collection and the patient-derived tumor xenograft model collection (PDX) consist of about 21,300 and 500 samples, respectively, for which gene expression profiles, generated with Affymetrix technology (Human Genome U133 plus 2.0 array), are available. The samples of the collection were internally annotated with controlled vocabulary for pathology, histology, and primary site using the COSMIC Classification System (*Forbes et al., 2008*), and were submitted for sensitivity prediction (*Figure 6—source data 2*). The associated GeneChip data were gathered from both public (GEO, ArrayExpress) and internal sources, and RMA normalized as described above. The naïve Bayes model, used to predict NVP-CFC218 sensitivity, was trained on the 251 CCLE expression data restricted to the 13-gene signature features.

PDX predictions of NVP-CFC218 sensitivity under the 13-gene signature were undertaken almost as described above. However, to account for the latest NVP-CFC218 pharmacological characterization on cell lines and to take advantage of a larger training data set, the naïve Bayes classifier was trained on a wider cell line compendium (634 cell lines, 77 and 557 of which being NVP-CFC218-sensitive and NVP-CFC218-insensitive, respectively). Moreover, the naïve Bayes-predictive model probability threshold, above which a xenograft tumor model is predicted as sensitive to p53–HDM2 inhibition, was also further set to 0.2 (optimizing for PPV/Precision at the cost of Sensitivity/Recall).

## In vivo studies in mice

All animal studies were conducted in accordance to procedures covered by permit number 1975 issued by the Kantonales Veterinäramt Basel-Stadt and strictly adhered to the Eidgenössisches Tierschutzgesetz and the Eidgenössische Tierschutzverordnung. All animals were allowed to adapt for 4 days and housed in a pathogen-controlled environment (5 mice/Type III cage) with access to food and water ad libitum and were identified with transponders.

### SJSA-1-derived tumor model

$3.0 \times 10^6$ SJSA-1 osteosarcoma cells were implanted subcutaneously in nude mice (Harlan, Germany). Treatment was initiated when tumors reached 150–200 mm$^3$ volume. Tumor response is

reported as percentage change in tumor volume at last day of treatment relative to start of treatment. The body-weight is reported as the percentage change relative to day 0.

## Patient-derived tumor xenografts (PDX) models

Surgical tumor tissues from treatment-naive cancer patients were implanted in the right flank of Harlan nude mice. All samples were anonymized and obtained with informed consent and under the approval of the institutional review boards of the tissue providers and Novartis. PDX models were histologically characterized and genetically profiled using various technology platforms after serial passages in mice. External diagnosis was independently confirmed by in-house pathologists. The p53 mutation status was determined for each PDX model by both RNA and DNA deep sequencing technologies. Efficacy studies and tumor response were measured as above. The cut-off used for sensitivity to NVP-CGM097 was based on RECIST adapted to tumor volume: models considered to be sensitive either displayed a complete response (full regression) or a partial response (>50% decrease in tumor volume) or a stable disease (between 50% decrease and 35% increase in tumor volume). In vivo models showing a progressive disease (>35% increase in tumor volume) were considered as non-responsive to NVP-CGM097 treatment. The sensitivity call was made at the maximum effect time-point during the treatment period, to avoid a call that could be due, for example, to the appearance of resistance mechanism(s) following treatment with NVP-CGM097.

## Pharmacodynamic marker evaluation in tumor

### p21 (CDKN1A) mRNA expression

Total RNA was purified using the QIAshredder and RNeasy Mini Kit (Qiagen, Valencia, CA). The quantitative reverse transcriptase polymerase chain reaction (qRT-PCR) for p21 was performed in triplicates using the One-Step RT qPCR Master Mix Plus (Eurogentec, Seraing, Belgium), with either control primers and primers for human p21 (Hs00355782_m1, Applied Biosystems, Carlsbad, CA), namely TaqMan Gene Expression kit assays (20x probe dye FAM [or VIC]-TAMRA [or MGB]; Applied Biosystems).

### p21 protein expression

ZeptoMARK chips, reagents, and protocols (Zeptosens, Bayer Technology Services, Germany) were used for the reverse phase protein microarray measurements (RPPA). Total protein was extracted from tumor samples with NP40-based lysis buffer. The primary anti-p21 antibody was from CST, Danvers, MA (#2947) and the secondary Alexa Fluor 647-labeled antibody was from Invitrogen (#Z25305). Fluorescence imaging of assay signals was performed with the ZeptoREADER instrument (Zeptosens). Microarray images were analyzed with the ZeptoVIEW Pro 2.0 software (Zeptosens).

## Immunochemistry

Immunohistochemistry has been performed on a Ventana Discovery XT automated immunostainer. SJSA-1 xenograft tumors were collected and a 3- to 4-mm slice was cut out of the middle of the tumor, fixed in neutral buffered formalin and embedded in paraffin. 3 μm sections were processed for immunohistochemistry (IHC). Primary antibodies used were mouse monoclonal anti-p21 antibody (#M7202, Dako, Carpenteria, CA) and mouse monoclonal anti-HDM2 antibody (#965, Santa-Cruz Biotechnology, Santa-Cruz, CA). Sections were subsequently stained using the labeled polymer system Simple Stain Mouse MAX PO (M) from the N-Histofine Mousestain Kit (Nichirei Bioscience Inc., Japan) and DAB substrate from the DABMap Kit omitting the SA-HRP solution (Ventana/Roche Diagnostics, Mannheim, Germany). Counterstaining of sections was done using hematoxylin (Ventana/Roche Diagnostics).

## Acknowledgements

The authors thank Geneviève Albrecht, Astrid Pornon, Simon Baer, and Monika Grueninger for technical assistance with cellular assays, Frédéric Baysang for technical assistance with biochemical assays, Dario Sterker for technical help with PK/PD relationship studies, Marjorie Berger, Ramona Rebmann, Francesca Santacroce, and Sonja Tobler for technical assistance with in vivo profiling,

Louise Barys for her help with the genomic and genetic profiling and Chris Wilson for running the high-throughput cell viability assay.

## Additional information

### Competing interests

SJ, SG, SF, HB, MI, TV, MM, SR, DAG, CR, MRJ, MW, JK, PF, FG, PH, KM, JW, EH, FH: Employee of Novartis Institutes for BioMedical Research. WRS: Was an employee of Novartis Institutes for Bio-Medical Research and is now an employee of Peptidream Inc. and has ownership interest (including patents) in Peptidream Inc. DGP: Employee of Novartis Institutes for BioMedical Research. Holds the position of VP/Global Head of Oncology in Novartis Institutes for BioMedical Research and has ownership interest (including patents) in Novartis Pharmaceuticals.

### Funding

| Funder | Author |
| --- | --- |
| Novartis | Sébastien Jeay |
| | Swann Gaulis |
| | Stéphane Ferretti |
| | Hans Bitter |
| | Moriko Ito |
| | Thérèse Valat |
| | Masato Murakami |
| | Stephan Ruetz |
| | Daniel A Guthy |
| | Caroline Rynn |
| | Michael R Jensen |
| | Marion Wiesmann |
| | Joerg Kallen |
| | Pascal Furet |
| | François Gessier |
| | Philipp Holzer |
| | Keiichi Masuya |
| | Jens Würthner |
| | Ensar Halilovic |
| | Francesco Hofmann |
| | William R Sellers |
| | Diana Graus Porta |

All research costs were covered by Novartis AG. No additional external funding was received for this study. The funder had no role in study design, data collection and interpretation, or the decision to submit the work for publication.

### Author contributions

SJ, DGP, Conception and design, Acquisition of data, Analysis and interpretation of data, Drafting or revising the article; SG, SF, HB, Acquisition of data, Analysis and interpretation of data, Drafting or revising the article; MI, TV, MM, SR, DAG, CR, Acquisition of data, Analysis and interpretation of data; MRJ, MW, JW, EH, Analysis and interpretation of data, Drafting or revising the article; JK, PF, FG, PH, KM, Acquisition of data, Contributed unpublished essential data or reagents; FH, WRS, Conception and design, Drafting or revising the article

### Ethics

Animal experimentation: All animal studies were conducted in accordance to procedures covered by permit number 1975 issued by the Kantonales Veterinäramt Basel-Stadt and strictly adhered to the Eidgenössisches Tierschutzgesetz and the Eidgenössische Tierschutzverordnung. All animals were allowed to adapt for 4 days and housed in a pathogen-controlled environment (5 mice/Type III cage) with access to food and water ad libitum and were identified with transponders.

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
