## [Decision Letter]

Thank you for sending your work entitled "A distinct p53 target gene set predicts for response to NVP-CGM097, a novel and selective p53-HDM2 inhibitor" for consideration at *eLife*. Your article has been favorably evaluated by Sean Morrison (Senior editor) and three reviewers, one of whom is a member of our Board of Reviewing Editors.

The Reviewing editor and the other reviewers discussed their comments before we reached this decision, and the Reviewing editor has assembled the following comments to help you prepare a revised submission.

This manuscript by Jeay et al. reports the discovery of a 13-gene signature that can predict the sensitivity of tumor cell lines and patient-derived tumor xenografts to small molecule inhibitors of the p53-MDM2 interaction with great accuracy, sensitivity and specificity. The manuscript is conceptually simple yet of very high impact.

The current manuscript not only reports the discovery of two novel inhibitors of the p53-MDM2 interaction, but also describes an experimental tour-de-force to identify the much-needed biomarker. These efforts lead the authors to identify a 13-gene signature that can predict the cellular response in vitro and in vivo, including patient-derived xenografts, with great accuracy, sensitivity and specificity. Remarkably, the gene signature is composed of 13 known direct transcriptional targets of p53 whose expression is strongly elevated in cell lines and tumors prior to MDM2 inhibition. This indicates that the sensitive cell lines harbor a more transcriptionally active form of p53 that 'primes' a fraction of its transcriptional program in proliferating cancer cells. This finding counters the prevalent notion that sensitivity is defined by MDM2 amplification, which would result in very low levels of active p53.

Although the paper does not make great mechanistic insights into p53 biology, its broad applicability makes it of very high impact. If this biomarker works in the clinic, it is likely to profoundly change the course of the current clinical trials and future use of MDM2 inhibitors in cancer therapy. Accordingly, the reviewers agreed on recommending resubmission of a revised manuscript addressing the following major points:

1) It seems like p53^WT^ (mutational status) is a good predictive marker for this compound, why is this information not taken into account when building the naive Bayes classifier? Can the prediction improve when p53 mutational status is taken into account as part of the gene signature? Why not try to build a gene signature based on p53 wild type that is sensitive or insensitive to this compound? It seems like the validation steps of the gene signature are built on the logic that p53 wild type could be used as a feature, as the independent in vitro validation was only performed on the p53^WT^ cell lines, and the PDX models were enriched with p53^WT^ models.

2) The gene selection method based on the Wilcoxon test seems to be a little bit primitive. There are better and more advanced feature selection methods (like lasso type or machine learning approaches). Have the authors tried different approaches and come to the conclusions with the same set of genes? Have the authors tested with different machine learning approaches (e.g. SVM?)

3) The fold-change of the 13 selected genes seems to be modest, some of them even less than 2 fold. Can these changes be validated by other experimental approaches like RT-PCR? It seems like the training of the classifier is based on the full *~*52,000 probe sets, not genes. Have the authors found multiple probe sets representing the same genes ranked high in the feature selection step? This may strengthen the inclusion of these genes.

4) How is this gene signature going to translate into a biomarker test in a clinical trial? The hardest part is the RMA normalization of a new sample. In the validation steps, both in vitro and in vivo gene expression profiles were normalized as a batch, and the expression values are normalized with the training set? This is unclear, and how will the authors envision normalizing patient samples? As in a clinical trial, patients will be tested individually, not by batch, therefore, getting the gene expression values normalized to the same scale with the training model needs careful consideration. This is not clearly described in the text.

5) Finally, in the training and testing in vitro cell lines, many of the sensitive cell lines are from the hematopoetic cell lineages, but none of this cell lineage is represented in the PDX Models. How would this affect the predictions of the classifier?

---

## [Author Response]

*1) It seems like p53*^*WT*^*(mutational status) is a good predictive marker for this compound, why is this information not taken into account when building the naive Bayes classifier? Can the prediction improve when p53 mutational status is taken into account as part of the gene signature? Why not try to build a gene signature based on p53 wild type that is sensitive or insensitive to this compound? It seems like the validation steps of the gene signature are built on the logic that p53 wild type could be used as a feature, as the independent in vitro validation was only performed on the p53*^*WT*^*cell lines, and the PDX models were enriched with p53*^*WT*^*models*.

As indicated in the manuscript, the evaluation of all feature types including gene expression, copy number and mutation yielded the *TP53* mutation status as being the top feature associated with response to NVP-CFC218, followed by the 13 p53 target-gene expression features. We compared the performance of the gene signature alone and with the *TP53* mutation feature. As shown in Figure 7, the performances as judged by ROC analyses were found similar whether *TP53* mutation was integrated into the prediction (as determined either by all exon sequencing [blue curve] or only exon 5-8 sequencing [green curve]) or not (red curve). This finding supports the hypothesis that the presence of 13-gene signature already reflects the p53 wild-type status, but also the p53 pathway functional activity and then the inclusion of p53 mutation status does not provide significant new discriminatory information.

Author response image 1.**DOI:**
http://dx.doi.org/10.7554/eLife.06498.019

Building a gene signature based on p53 wild-type was tried as well but was found to be mainly limited by the number of p53 wild-type cell lines that are available. As shown in Figure 1, a total of 113 p53 wild-type cells are available, of which only 43 are sensitive and 70 are insensitive to NVP-CFC218. For this reason, this approach lost a lot of power which greatly impacted the significance of the feature differences.

The gene signature and associated predictive model were built independently of the p53 mutation status, by comparing the chemical sensitivities of both p53 wild-type and mutant cell lines. As discussed above, the inclusion of p53 mutation status into the predictions did not increase its performance. Based on the mechanism of action of the molecules and on the results shown in Figure 1, p53 wild-type status is required for sensitivity. It was then important to test our hypothesis in already pre-selected p53 wild-type cell lines, especially because in a clinical setting, only pre-selected p53 wild-type patients are being treated with NVP-CGM097. On the contrary, the in vivo validation in PDX was primarily performed independently of the p53 mutation status. The tumor models were stratified according to the *TP53* mutation status in a retrospective manner, which allowed us to determine the performances of the gene signature in all 55 models (Figure 6 and [Supplementary-material SD4-data]), as well as in the p53 wild-type models only, as shown in Figure 6.

*2) The gene selection method based on the Wilcoxon test seems to be a little bit primitive. There are better and more advanced feature selection methods (like lasso type or machine learning approaches)*. *Have the authors tried different approaches and come to the conclusions with the same set of genes? Have the authors tested with different machine learning approaches (e.g. SVM?)*

The choice of the Wilcoxon rank-sum test was based on several factors. First, we have performed extensive mining of the CCLE using methods such as lasso, ridge regression, elastic net, or SVMs and have generally found similar results across the classification algorithms. To this aim, we employed these methods and found that the majority of the candidate features overlapped with what was found using the Wilcoxon test. Second, the Wilcoxon test is simple and conservative, two appealing properties for our purposes. Third, the method is quite scalable to the type of analysis we wanted to perform, i.e. gradually increasing the number of gene features to set the optimal classification under the most parsimonious set of genes. This analytical approach in combination with naïve Bayes probabilistic modeling is a machine learning approach with embedded feature selection, which is in our opinion an appropriate classification strategy.

*3) The fold-change of the 13 selected genes seems to be modest, some of them even less than 2 fold. Can these changes be validated by other experimental approaches like RT-PCR? It seems like the training of the classifier is based on the full ~52,000 probe sets, not genes. Have the authors found multiple probe sets representing the same genes ranked high in the feature selection step? This may strengthen the inclusion of these genes*.

Indeed the fold changes are modest, nonetheless highly statistically significant. A likely explanation could come from the Affymetrix GeneChip© technology together with the RMA summarization method that was used for the gene expression measurements. Such a protocol is known to compress fold changes. Even if a RT-PCR quantification of the transcript expression is expected to reach about 10-20% higher fold changes, this has not been verified on this data set. However, the nanostring quantification of the 13 genes making signature became available in the meantime for most of the cell lines. As expected, the fold changes were found to remain within the same range, even if slightly higher than those from the microarrays. The RMA summarization method is based on the Michigan alternative CDF that summarizes the CEL data at the Entrez gene level. Consequently we did not try to integrate the multi-probe set representations in the analysis to strengthen the differential expression of some genes. We have now added a paragraph in the Material and methods section entitled 'Affymetrix GeneChip© RMA normalization' to better clarify the alternate CDF summarization in the manuscript.

*4) How is this gene signature going to translate into a biomarker test in a clinical trial? The hardest part is the RMA normalization of a new sample. In the validation steps, both in vitro and in vivo gene expression profiles were normalized as a batch, and the expression values are normalized with the training set? This is unclear, and how will the authors envision normalizing patient samples? As in a clinical trial, patients will be tested individually, not by batch, therefore, getting the gene expression values normalized to the same scale with the training model needs careful consideration. This is not clearly described in the text*.

We do not foresee major difficulties to the prediction of independent samples using the gene signature, provided the same expression quantification technology is used (Affymetrix GeneChip©). To specifically discuss the RMA normalization step, all samples from the training as well as those from the test sets were not normalized all together as a batch, but with respect to an independent external data set consisting of both cancerous and normal samples of various lineage origins. Thus, any independent test sample can be individually RMA normalized to this reference batch. We have now added a paragraph in the Material and methods section entitled 'Affymetrix GeneChip© RMA normalization' to better clarify the RMA normalization step in the manuscript.

*5) Finally, in the training and testing in vitro cell lines, many of the sensitive cell lines are from the hematopoetic cell lineages, but none of this cell lineage is represented in the PDX Models. How would this affect the predictions of the classifier*?

The cell lines were picked up randomly with regards to lineage in the training and testing sets. The hematopoietic lineage was well represented in the training set (16%) even though the hematopoietic lineage (or any other lineage) as a feature was not found to be significantly over-represented in the sensitive cell lines globally. Therefore we did not anticipate that the hematopoietic lineage composition of the sensitive cell line set would interfere with or bias the feature selection or the naïve Bayes predictions. Because of the cell line availability, the hematopoietic lineage was also well represented in the testing set (27%), and indeed, many of them proved to be sensitive to both MDM2 inhibitors. On the contrary, no PDX models of the hematopoietic lineage are available in our collection and the lineage composition of the predictive model couldn't be reproduced in the in vivo validation.